# BEHAVIOR MODULE IN NEURAL NETWORKS

## ABSTRACT

Prefrontal cortex (PFC) is a part of the brain which is responsible for behavior repertoire. Inspired by PFC functionality and connectivity, as well as human behavior formation process, we propose a novel modular architecture of neural networks with a Behavioral Module (BM) and corresponding end-to-end training strategy. This approach allows efficient learning of behaviors and preferences representation. This property is particularly useful for user modeling (as for dialog agents) and recommendation tasks, as allows learning personalized representations of different user states. In the experiment with video games playing, the results show that the proposed method allows separation of main task's objectives and behaviors between different BMs. The experiments also show network extendability through independent learning of new behavior patterns. Moreover, we demonstrate a strategy for an efficient transfer of newly learned BMs to unseen tasks.

## 1 INTRODUCTION

Humans are highly intelligent species and are capable of solving a large variety of compound and open-ended tasks. The performance on those tasks often varies depending on a number of factors. In this work, we group them into two main categories: *Strategy* and *Behaviour*. The first group contains all the factors leading to the achievement of a defined set of goals. On the other hand, Behaviour is responsible for all the factors not directly linked to the goals and having no significant effect on them. Examples of such factors can be current sentiment status or the unique personality and preferences that affect the way an individual makes decisions. Existing Deep Networks have been focused on learning of a Strategy component. This was achieved by optimization of a model for defined sets of goals, also the goal might be decomposed into sub-goals first, as in FeUdal Networks (Vezhnevets et al., 2017) or Policy Sketches approach (Andreas et al., 2017). Behavior component, in turn, obtained much less attention from the DL community. Although some works have been conducted on the identification of Behavior component in the input, such as works in emotion recognition (Kahou et al., 2016; Han et al., 2014; Levi & Hassner, 2015). To the best of our knowledge, there was no previous research on incorporation on Behavior Component or Behavior Representation in Deep Networks before. Modeling Behaviour along with Strategy component is an important step to mimicking a real human behavior and creation of robust Human-Computer Interaction systems, such as a dialog agent, social robot or recommendation system.

The early work of artificial neural networks was inspired by brain structure (Fukushima, 1980; LeCun & Bengio, 1995), and the convolution operation and hierarchical layer design found in the network designed for visual analytic are inspired by visual cortex (Fukushima, 1980; LeCun & Bengio, 1995). In this work, we again seek inspiration from the human brain architecture. In the neuroscience studies, the prefrontal cortex (PFC) is the region of the brain responsible for the behavioral repertoire of animals (Miller & Cohen, 2001). Similar to the connectivity of the brain cortex (as shown in Figure 1), we hypothesize that a behavior can be modeled as a standalone module within the deep network architecture. Thus, in this work, we introduce a general purpose modular architecture of deep networks with a Behavioural Module (BM) focusing on impersonating the functionality of PFC.

Apart from mimicking the PFC connectivity in our model, we also borrow the model training strategy from human behavior formation process. As we are trying to mimic the functionality of a human brain we approached the problem from the perspective of Reinforcement Learning. This approach also aligns with the process of unique personality development. According to Depue et al. (1994) and Depue & Collins (1999) unique personality can be explained by different dopamine functions caused by genetic influence. These differences are also a reason for different Positive Emotionality (PE)

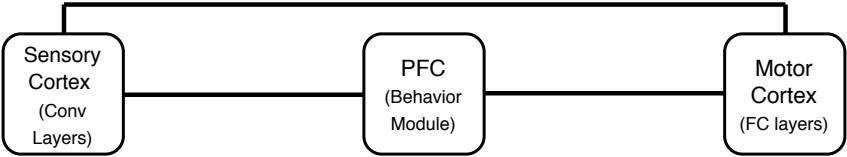

Figure 1: Abstract illustration of the prefrontal cortex (PFC) connections of the brain (Miller & Cohen, 2001) and corresponding parts of the proposed model.

patterns (sensitivity to reward stimuli), which are in turn a significant factor in behavior formation process (Depue & Collins, 1999). Inspired by named biological processes we introduce extra positive rewards (referring to positive-stimuli or dopamine release, higher the reward referring to higher sensitivity) to encourage specific actions and provoke the development of specific behavioral patterns in the trained agent.

To validate our method, we selected the challenging domain of classic Atari 2600 games (Bellemare et al., 2013), where the simulated environment allows an AI algorithm to learn game playing by repeatedly seek to understand the input space, objectives and solution. Based on this environment and an established agent (*i.e.* Deep Q-Network (DQN) (Mnih et al., 2015)), the behavior of the agent can be represented by preferences over different sets of actions. In other words, in the given setting, each *behaviour* is represented by a probability distribution over given action space. In real-world tasks, the extra-reward can be represented by the human satisfaction by taken action along with the correctness of the output (main reward).

Importantly, the effect of human behavior is not restricted to a single task and can be observed in various similar situations. Although it is difficult to correlate the effect of human behavior on completely different tasks, it is often easier to observe akin patterns in similar domains and problems. To verify this, we study two BM transfer strategies to transfer a set of newly learned BMs across different tasks. As a human PFC is responsible for behavior patterns in a variety of tasks, we also aim to achieve a zero-shot transfer of learned modules across different tasks.

The contributions of our work are as follow:

- We propose a novel modular architecture with behavior module and a learning method for the separation of behavior from a strategy component.
- We provide a 0-shot transfer strategy for newly learned behaviors to previously unseen tasks. The proposed approach ensures easy extendability of the model to new behaviors and transferability of learned BMs.
- We demonstrate the effectiveness of our approach on video games domain. The experimental results show good separation of behavior with different BMs, as well as promising results when transfer learned BMs to new tasks. Along with that, we study the effects of different hyper-parameters on the behavior separation process.

## 2 RELATED WORK

Task separation is an important yet relatively unexplored topic in deep learning. In 1989, Rueckl et al. (1989) explored this idea by simulating a simplified primate visual cortex by separation a network into two parts, responsible for shape classification task and shape localization on a binary image, respectively The topic was further studied in Jacobs (1990); Jacobs et al. (1991a;b), however, due to the limitations in computational resources at that time, it has not gotten much advancement.

Recently, number researchers have revisited the idea of *task separation* and *modular networks* with evolutionary algorithms. So in 2005, Stanley et al. (2005) and Reeder et al. (2008) applied neuroevolution algorithms to evolve predefined modules responsible for the problem subtasks, where improved performance was reported when compared against monolithic architectures. Schrum & Miikkulainen (2009; 2012) proposed a neuroevolution approach to develop a multi-modular network capable of learning different agent behaviors. The module structure and the number of modules in the network were evolved in the training process. Although the multi-module architecture achieved better performance, it has not achieved separation of the tasks among the modules. A number of evolved

modules appeared redundant and not used in the test phase, while others have used shared neurons. Moreover, the architecture was fixed once learned and did not assume changes in the structure. The same approach with modifications in mutation strategy '(Schrum & Miikkulainen, 2014), genome encoding (Schrum et al., 2016) and task complexity (Schrum & Miikkulainen, 2014; 2015), but has not achieved significant performance.

In 2016, Braylan et al. (2016) proposed to use a coevolutionary algorithm for domain transfer problem to avoid training from the scratch. It first independently learns a pool of networks on different Atari2600 games, During the transfer phase, the networks were frozen and used as a 'building blocks' in a new network while combined with newly evolved neurons. In 2017, Fernando et al. (2017) introduced PathNet to address the task-transfer module on the example of Atari2600 games. PathNet has a fixed size architecture (L layers by N modules), where each module was represented by either convolutional or fully-connected block. During the training phase, authors applied the tournament genetic algorithm to learn active paths between modules along with the weights. Once the task was learned, active modules and paths were frozen and the new task could start learning a new path. Recently proposed FeUdal Networks architecture (Vezhnevets et al., 2017), also proposed a Modular design for Reinforcement Learning problems with sub-goals. In this work authors use Manager and Worker modules for learning abstract goals and primitive actions respectively. FeUdal networks are designed to tackle environments with long-term credit assignment and sparse reward signals. The modules in the named architecture are not transferable and designed to learn different time-span goal embeddings.

Andreas et al. (2016) proposed the Neural Module Network for Visual Question Answering (VQA) task. It consists of separate modules responsible for different tasks (*e.g.* Find, Transform, Combine, Describe and Measure modules), which could be combined in different ways depending on the network input. A similar dynamic architecture was proposed and applied to robot manipulator task (Devin et al., 2017). The model was end-to-end trained and consisted of two modules (*i.e.* robot-specific and task-specific) and achieved good performance on a zero-shot learning task. The Modular Neural Network was also applied in Reinforcement Learning task in a robotics environment (Andreas et al., 2017). In this work, each module was responsible for a separate sub-task of the main task. However, the modules could be combined only in a sequential manner.

Most of the previous works focused on multi-task problems or problems with sub-goals where the modules were responsible for learning explicit sub-functionality directly affecting the model performance. Our approach is different in a sense, we learn a behavior module responsible for representation of user sentiment states or preferences not affecting the main goals. This approach leads to high adaptability of the network performance to new preferences or states of an end-user without retraining of the whole network, expandability of the network to future variations, removability of BMs in case of unknown preferences, as well as high-potential to transfer of the learned representations to unseen tasks. To the best of our knowledge, there are no similar approaches.

## 3 METHOD

The goal of our modular network is to introduce Behavior component into Deep Networks, ensure separation of the behavior and main task functionalities into different components of the network and provide a strategy for an efficient transfer of learned behaviors. Our model has three main parts (1) The **Main Network** is responsible for the main task (strategy component) functionality, (2) a replaceable/removable **Behavior Module** (BM) encodes the agent behavior and separate it from the main network, and (3) the **Discriminator** is used during the transfer stage and helps to learn similar feature representations among different tasks. An overview of the proposed network architecture is shown in Figure 2. In the given architecture Convolutional layers correspond to (Visual) Sensory cortex, Fully-Connected layers of the Main Network to the Motor Cortex and Behaviour Module to PFC from Figure 1.

### 3.1 MAIN NETWORK

In this work, we adopt the deep Q-Network (DQN) with target network and memory replay (Mnih et al., 2015) to solve the main task (denoted as *main network*). DQN has reported good performance

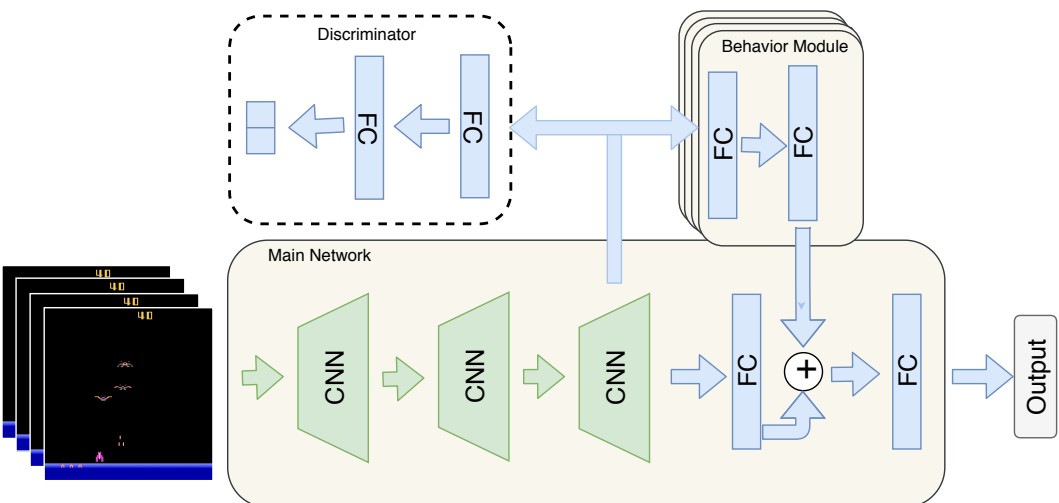

Figure 2: Overview of the proposed modular network architecture with Behavioral Module (BM) and main network represented by DQN (Mnih et al., 2015). The Discriminator is only used for adversarial BM transfer.

on Deep Reinforcement Learning task and achieved human-level performance on a number of Atari2600 games (Bellemare et al., 2013).

The DQN has a fairly simple network structure, which consists of 3 consecutive convolutional layers followed by two fully-connected (fc) layers (see Figure 2). All the layers, except the last one, use ReLU activation functions. The network output is represented by the set of expected future discounted rewards for each of the available actions. The output obeys Bellman-equation and Root-Mean-Square Error is applied as a loss function ($\mathcal{L}_m$) during the training phase.

## 3.2 BEHAVIOR SEPARATION

In this work, we are interested in the separation of the behavioral component from the main task functionality. Specifically, we design a network where the behavior is modeled with a replaceable and removable module, which is denoted as **Behavioral Module** (BM). The BM is supposed to have no significant effect on the performance on the main task.

The BM is modeled as two fc layers with the first layer having ReLU activation function and the second layer having linear activation. The proposed BM input is the output from the last convolutional layer of the main network, while its output is directly fed to the first fc layer of the main network. This architecture follows the PFC connectivity pathways described in Figure 1. The forward pass of the network is represented by the following equations:

$$l_1 = f_1(\boldsymbol{I} \circledast \boldsymbol{W}_1 + \boldsymbol{b}_1)$$
$$l_2 = f_2(l_1 \circledast \boldsymbol{W}_2 + \boldsymbol{b}_2)$$

$$l_3 = f_3(l_2 \circledast \boldsymbol{W}_3 + \boldsymbol{b}_3)$$
$$l_{b1} = f_{b1}(l_3 \cdot \boldsymbol{W}_{bj1} + \boldsymbol{b}_{bj1})$$
$$l_{b2} = l_{b1} \cdot \boldsymbol{W}_{bj2} + \boldsymbol{b}_{bj2}$$

$$l_4 = f_4(l_3 \cdot \boldsymbol{W}_4 + \boldsymbol{b}_4 + l_{b2})$$
$$l_5 = l_4 \cdot \boldsymbol{W}_5 + \boldsymbol{b}_5,$$

where $\boldsymbol{I}$ is the network input, $j$ is the index of the current behaviour, $l_i$ is the output of the $i$-th layer, $l_{bi}$ is the output of i-th layer of a BM, $f_i$ is the activation function at the $i$-th layer, and $\circledast$ denotes 2d convolution operation. The Main Network contains layers from $l_1$ to $l_5$. Note that $l_b$ becomes zero vector if no behavior is required or enforced.

The summation operator at the layer 4 ensures the influence of BM can be easily removed from the main network ($l_b$ is zeros in this case). It also minimizes the effects of BM on the gradients flow during the backpropagation stage. The training is conducted in an end-to-end manner as presented in Algorithm 1.

In our approach, the introduction of BM does not require additional loss function and the loss is directly incorporated into the main network loss ($\mathcal{L}_m$). To do this we introduce additional rewards for desired behaviors of the agent, similar to PE effect on human behavior formation process. In our

---

**Algorithm 1** Behavior Separation

---
1: **procedure** TRAINBEHAVIORNET
2:     **input:** $N$ - number of training iterations, $R$ - set of replay buffers,
3:     **input:** $\theta$- network parameters, $\theta'$- target network parameters,
4:     **input:** $B$ - set of behavior blocks, $N'$ - target network update frequency
5:     $i \leftarrow$ random(1,size($B$)); $\theta' \leftarrow \theta$
6:     **for** t in range(0,N) **do**
7:         play game with i-th bevaior module
8:         store new data to $R$[i]
9:         $j \leftarrow$ random(1,size($B$))
10:         sample training batch from $R$[j]
11:         Optimize $\theta$ and $B$[j]
12:         Replace $\theta' \leftarrow \theta$ every $N$' steps
13:         **if** game finished **then**
14:             $i \leftarrow$ random(1,size($B$))

---

setting *behavior* is defined by agent's preference to play specific actions. Thus, each preferred action played was rewarded with an extra action reward. The action reward is subject to the game of interest and its designing process will be described in the Experiment section.

### 3.3 BEHAVIOR TRANSFER

One of the advantages of network modularization is to allow the learned BMs to be transferred to a different main network with minimal or no network re-training. This property is useful for knowledge sharing among a group of models in problems with a variety of possible implementations, changing environments and open-ended tasks. Once task objectives have changed or new behaviors were developed in another model, the target model can just apply a new module without any updates or training of the main network. This property allows easy extension of a learned model and knowledge share across different models without any additional training.

The learned BMs from the previous section is used during the transfer phase. In this work, we consider two approaches, namely *fine-tuning* and *adversarial transfer*. The first approach uses a source task model and fine-tunes it for a new target task, where BMs are kept unchanged.

In the adversarial transfer approach, we introduce a discriminator network (as shown in Figure 2) , which enforces the convolutional layers to learn features similar to features of the source task. To do so, we adopt the domain-adversarial training (Ganin et al., 2016). In this case, the discriminator network has 2 fully-connected layers with Relu and Softmax non-linearity functions, which tries to classify output of the last convolutional layers as being from the source or target task. Different from the original paper, we minimize the softmax loss at the discriminator output and flip the gradient sign at the convolutional layers. The weights update can be formulated as follows:

$$\theta_{t+1}^d = \theta_t^d - \beta \frac{\partial \mathcal{L}_a}{\partial \theta^d} \qquad \theta_{t+1}^c = \theta_t^c + \beta \frac{\partial \mathcal{L}_a}{\partial \theta^c}$$

where $\theta_t^{dj}$ are the parameters of the discriminator at timestep t, $\theta_t^{aj}$ are the parameters of the convolutional layers at timestep t, $\beta$ is the parameter as described by Ganin et al. (2016), $\mathcal{L}_a$ is the classification loss of the descriminator.

## 4 EXPERIMENT

In this section, we delineate the experiments that focus on two main aspects of this work: (1) the separation of agent's behavior from the main task, and (2) cross-task transfer of learned behaviors. In order to demonstrate the flexibility and extendability of the proposed network, we also considered zero-shot setting so that an end-user will not require additional training for the case of behavior module transfer.

**Environment:** We evaluate the proposed novel modular architecture on the classic Atari 2600 games (Bellemare et al., 2013). The main reason is that video games provide a controlled environment,

Table 1: Game scores and Behavior Distance score (in the brackets) achieved with vanilla DQN model (Mnih et al., 2015) and our proposed model with optimal parameters.

|                | Pong         | Space Invaders | Demon Attack  | Breakout      |
|----------------|--------------|----------------|---------------|---------------|
| DQN            | 21.0 (-)     | 726.2 (-)      | 1564.8 (-)    | 115.9 (-)     |
| Our (BM0)      | 21.0 (-)     | 727.1 (-)      | 2658.8 (-)    | 127.5 (-)     |
| Ours (Stage 1) | 21.0 (0.96)  | 617.3 (0.86)   | 2128.1 (0.83) | 124.0 (0.87)  |
| Ours (Stage 2) | 21.0 (0.95)  | 624.6 (0.88)   | 1893.4 (0.85) | 117.1 (0.88)  |

where it is easier to control agent behavior by representing it with distribution over available actions. In addition, Atari 2600 emulator does not require data collection and labeling, yet it provides a wide range of tasks in terms of different games. The loss function used to encourage the learning of a specific behavior is described in the next section. In this work, we evaluate our architecture on four games, namely *Pong*, *Space Invaders*, *Demon Attack* and *Breakout*, which consist of four available actions, namely *No action*, *Fire*, *Up/Right*, and *Down/Left*.

**Data pre-processing:** The input of the network is a stack of 4 game-frames. Each frame was converted into a gray-scale image and resized to $84 \times 84$ pixels. As the consecutive game frame contain similar information, we sample one game frame on every fourth frame.

**Behavior:** We design 8 possible behaviors to simulate various action sets. BM1, BM2, BM3, and BM4 encouraged playing a single action (*i.e.* No action, Fire, Up/Right or Down/Left); BM5 and BM6 stimulated reasonable pairs of actions (*i.e.* Up and Down (or Right and Left), and Fire and No action (or Fire and Right)); BM7 and BM8 encouraged playing sets of 3 actions represented by BM5 and an additional action. Additionally, we tested the effect of zero-behavior (*i.e.* BM0) presence during the training stage. In other words, no actions were stimulated and BM is not applied.

**Training:** To train the proposed network (*i.e.* main network and BMs), we used the standard Q-learning algorithm and the corresponding loss function using Bellman equation (Mnih et al., 2013). The training used a combined reward represented by the sum of game score and individual action rewards. The magnitude of the additional reward was represented by an average reward per frame of the game. All the rewards obtained during the game were clipped at 1 and -1 (Mnih et al., 2015).

**Evaluation Metrics:** We evaluate the proposed models using two metrics. The first metric focus on the game play performance. As each game has different reward scales, we compute the mean of game scores ratios achieved by the proposed modular network and the Vanilla DQN model (Table 1). We refer to this metric as *Average Main Score Ratio* (AMSR). If AMSR is equal to 1, it means the trained model with BM performs equally well as the Vanilla DQN model. Similarly, AMSR higher than 1 indicate our proposed model perform better than the Vanilla DQN, or worst if it is lower than 1. Thus, AMSR that is close to or more than 1 would indicate our modular network is comparable to baseline.

The second metric reflects the capability of the proposed modular network in term of modeling the desired behavior. To do this, we define the *Behavior Distance* (BD) by calculating the Bhattacharyya distance (Bhattacharyya, 1943) between the BMs' action distribution to an ideal target distribution. The target distribution is computed by divide 1 with the rewarded actions (*i.e.* BM5's target distribution is $[0.0, 0.0, 0.5, 0.5]$). In the ideal case, the BD of the learned network should be close to 1 as our training only encourages over certain actions set.

## 4.1 Experiment on Behavior Separation

This experiment aims to show that the behavior functionality can be separated from the main task and learned independently. To demonstrate that we conducted the training in two stages. During the Stage 1, we first trained the main network with five behaviors (*i.e.*, BM0 - BM2, BM4 - BM5 and BM8) using Algorithm 1. Given the trained network from Stage 1, Stage 2 focused on training of the remaining BMs (*i.e.* BM3, BM6, and BM7) while the main network was frozen. This includes behaviors stimulating 1, 2 and 3 actions, respectively.

Table 2: Effect of action reward magnitude, where $r$ is the average reward per frame size of trained network with vanilla DQN model.

| Action Reward | | AMSR | | BD | |
| --- | --- | --- | --- | --- | --- |
| | BM0 | Stage 1 | Stage 2 | Stage 1 | Stage 2 |
| 0.25r | 1.04 | 0.90 | 0.80 | 0.74 | 0.79 |
| 0.50r | 1.04 | 0.89 | 0.88 | 0.83 | 0.85 |
| 0.75r | 1.21 | 0.95 | 0.82 | 0.88 | 0.87 |
| r | 0.93 | 0.88 | 0.85 | 0.87 | 0.89 |
| 2r | 0.65 | 0.67 | 0.67 | 0.94 | 0.78 |
| 5r | 0.42 | 0.49 | 0.46 | 0.97 | 0.82 |

Table 3: Effect of the number of layers in the Behavioural Module.

| Number of layers | | AMSR | | BD | |
| --- | --- | --- | --- | --- | --- |
| | BM0 | Stage 1 | Stage 2 | Stage 1 | Stage 2 |
| 1 | 1.04 | 0.89 | 0.88 | 0.83 | 0.85 |
| 2 | 1.25 | 1.18 | 1.06 | 0.83 | 0.84 |
| 3 | 1.26 | 1.18 | 1.00 | 0.82 | 0.85 |

**Effect of key parameters:** First of all, we studied the effect of the action reward magnitude on the training process. We started with estimation of average reward per frame value ($r$) for each game (without additional action rewards) and observed performance of our model with various action rewards (*i.e.* 0.25r, 0.5r, r, 2r, and 5r). Table 2 show that action reward magnitude directly affects the quality of learned behavior in both stages, where increasing the action reward above $r$ value leads to degradation of the main task's performance. Although additional reward magnitude selection depends on a desired main score and BD, we recommend the value equal to $r$ as it leads to the highest BD score during Stage 2, and as a result better functionality separation. Next, to see the effect of other parameters we set the value of the action reward to $0.5r$, so that we can observe an effect of the changes on the main reward, as well as the behavior pattern.

As the next step, we have studied the effect of the complexity of the BMs on the quality of the learned behaviors by trying a different number of layers. Also, we looked for a better separation of the Behavior component by studying the effects of dropout, BM0 and different learning rate for the Behavior module. According to the results (Table 3) use of 2 fully-connected layers resulted in a significant improvement on the main task score compared to 1-layer version. However, adding more layers did not result in a better performance. Similar effect demonstrated a higher BM learning rate compared to the main network (Table 4), while lower value leads to lower main scores. Finally implementing a Dropout layer for the BM and using BM0 resulted in a higher BD score during Stage 2 and main score during Stage 1.

**Results:** Taking into account hyper-parameter effect we have trained a final model with 2 layer BMs and 0.5 dropout layers, applying 2 times higher BM learning rate, BM0 and action reward equal to $r$. The trained model showed high main task scores compared to the vanilla DQN model, as well as high similarity of learned behaviors to ideal target distributions at Stage 1, as well as after separate training of BMs at Stage 2 (Table 1). Experiments also showed that removing the BMs does not lead to a performance degradation of the model on the main task. Importantly, the effect of the action reward magnitude directly correlated with agents preferences to play rewarded actions, which aligns with the PE effect in human behavior formation process. Thus, the development process of exact behavior pattern can be controlled through variations in action reward magnitude. Therefore, we conclude that the proposed model and training method allows a successful separation of the strategy (main task) and behavior functionalities and further network expansion.

Table 4: Effect of learning rate magnitude of the Bahavioural Module

| BM learning rate multiplier | AMSR | | | BD | |
|---|---|---|---|---|---|
| | BM0 | Stage 1 | Stage 2 | Stage 1 | Stage 2 |
| 0.5 | 0.81 | 0.84 | 0.80 | 0.81 | 0.85 |
| 1.0 | 1.04 | 0.89 | 0.88 | 0.83 | 0.85 |
| 2 | 1.10 | 0.91 | 0.89 | 0.82 | 0.85 |
| 3 | 1.05 | 0.90 | 0.90 | 0.83 | 0.85 |

Table 5: Zero-shot transfer results

| Approach | AMSR | | | BD | |
|---|---|---|---|---|---|
| | BM0 | Stage 1 | Stage 2 | Stage 1 | Stage 2 |
| No training | 1.20 | 1.07 | 0.97 | 0.88 | 0.67 |
| Fine-tuning | 0.93 | 0.92 | 0.93 | 0.85 | 0.75 |
| Adversarial | 0.95 | 1.03 | 0.95 | 0.86 | 0.81 |

## 4.2 EXPERIMENT ON BEHAVIOR TRANSFER

**Implementation details:** To achieve a zero-shot performance of the transferred modules, we aimed to achieve a similar feature representation of the target model to the source model. To achieve that we tested two approaches: *fine-tuning* and *adversarial transfer*. In the first case, we have fine-tuned the main network obtained in Stage 1 of Section 4.1 on a new game with frozen Stage 1 BMs, applied to every pair of games and following Algorithm 1. After that, we tested the performance on previously unseen Stage 2 BMs. In adversarial setting we follow the same procedure, but with the use of the Discriminator part (Figure 2). The performance was compared to the results of transferring Stage 2 BMs to the best model configuration after Stage 1 from Section 4.1.

**Results:** As it can be seen from the Table 5 even a simple zero-shot transfer of learned BMs based on fine-tuning results in a good performance of the model on unseen BMs. BM0 and Stage 1 behaviors achieved close performance to an original network. Although the BD score of zero-shot adversarial transfer is approximately 9% lower, the main task performance of transferred modules on an unseen task is close to a separately trained network. This fact shows that zero-shot transfer of separately learned BMs to unseen tasks results in slightly worse performance compared to the separately trained model. This leads to a conclusion that target performance of transferred BMs can be achieved through much less training compared to complete network retraining.

## 5 CONCLUSION

In this work, we have proposed a novel Modular Network architecture with Behavior Module, inspired by human brain Pre-Frontal Cortex connectivity. This approach demonstrated the successful separation of the Strategy and Behavior functionalities among different network components. This is particularly useful for network expandability through independent learning of new Behavior Modules. Adversarial 0-shot transfer approach showed high potential of the learned BMs to be transferred to unseen tasks. Experiments showed that learned behaviors are removable and do not degrade the performance of the network on the main task. This property allows the model to work in a general setting, when user preferences are unknown. The results also align with human behavior formation process. We also conducted an exhaustive study on the effect of hyper-parameters on behavior learning process. As a future work, we are planning to extend the work to other domains, such as style transfer, chat bots, and recommendation systems. Also, we will work on improving module transfer quality.

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

Table 6: Effect of the Dropout rate applied to the Behavioural Module.

| Dropout rate | AMSR | | | BD | |
|---|---|---|---|---|---|
| | Removed BMs | Stage 1 | Stage 2 | Stage 1 | Stage 2 |
| 0.0 | 1.04 | 0.89 | 0.88 | 0.83 | 0.85 |
| 0.1 | 1.00 | 0.92 | 0.86 | 0.86 | 0.85 |
| 0.3 | 1.03 | 0.92 | 0.87 | 0.86 | 0.87 |
| 0.5 | 1.19 | 0.94 | 1.01 | 0.86 | 0.87 |

Table 7: Effect of the training with/without BM0

| | AMSR | | | BD | |
|---|---|---|---|---|---|
| | BM0 | Stage 1 | Stage 2 | Stage 1 | Stage 2 |
| Without BM0 | 1.04 | 0.89 | 0.88 | 0.83 | 0.85 |
| With BM0 | 1.07 | 0.95 | 0.85 | 0.81 | 0.86 |

## A  APPENDIX

In this appendix, we show the details of our preliminary study on various key parameters. The experiments were conducted on the Behavior Separation task.

