# OpenReview forum: "BEHAVIOR MODULE IN NEURAL NETWORKS"
_ICLR.cc/2019/Conference_

### Official Review · AnonReviewer2 · 2018-11-02
**Interesting Paper, underwhelming experiments**

**Rating:** 4
**Confidence:** 5

**Review:**

This paper introduces a "behavior module" (BM) which is a small network that encodes preferences over actions and runs parallel to the fully connected layers of a policy. The paper shows this architecture working in Atari games, where the same policy can be used to achieve different action preferences over a game while still playing well. It also includes a thorough recap of past modular approaches.

The motivation for the BM is that we may want deep networks to be able to decompose "strategy" and "behavior", where behavior may influence decisions without affecting the performance. In this framework, the BM is trained on a reward of correctness + personalized “satisfaction”.

The experiments model behavior as preferences over how many actions to play simultaneously. The trained BMs can be transferred to new tasks without finetuning. The ideas here also have some similarity to the few shot learning literature.

Comments on the experiments:
1. The Table 2  do not show a smooth interpolation between reward scaling and AMSR vs BD. This is surprising because the performance on the game should be highest went it is weighted to the most. This indicates to me that the results are actually high variance, the 0.8 vs 0.88 in stage 2 of 0.25r vs 0.5r means that is probably at least +/- 0.08 standard deviation. Adding standard deviations to these numbers is important for scientific interpretability.
2. I expect some BMs should perform much better than others (as they have been defined by number of actions to play at once). I would like to see (maybe in the appendix) a table similar to table 2 for for individual BMs. I currently assume the numbers are averaged over all BMs.
3. Similarly, I would like to see the BD for BM0 (e.g., if a policy is not optimized for any behavior, how close does it get to the other behaviors on average). This is an important lower bound that we can compare the other BD to.
4. An obvious baseline missing is to directly weight the Q values of the action outputs  (instead of having an additional network)  by the designed behavior rewards. There is an optimal way to do this because of experimental choices.

Questions:
1.For BM2, you write " Up and Down (or Right and Left)" did you mean "Up and Right"? How can Up and Down be played at the same time?

Overall, this paper uses neuroscience to motivate a behavior module. However, the particular application and problem settings falls short of these abstract "behaviors". Currently, the results are just showing that RL optimizes whatever reward function is provided, and that architectural decomposition allows for transfer, which was already showed in (Devin 2017). An experiment which would better highlight the behavior part of the BM architecture is the following:
1. Collect datasets of demonstrations (e.g. on atari) from different humans.
2. Train a policy to accomplish the task (with RL)
3. Train BMs on each human to accomplish the task in the style of each human.
This would show that the BMs can capture actual behavior.

The dialog examples discussed in the abstract would also be very exciting.

In conclusion, I find the idea interesting, but the experiments do not show that this architecture can do anything new. The abstract and introduction discuss applications that would be much more convincing. I hope to see experiments with a more complex definition of "behavior" that cannot be handcoded into the Q function.

---

### Official Review · AnonReviewer1 · 2018-11-03
**Review for Paper BEHAVIOR MODULE IN NEURAL NETWORKS**

**Rating:** 3
**Confidence:** 5

**Review:**

The authors try to build a deep neural network model based on observations from the human brain Pre-Frontal Cortex connectivity. Based on a DQN network, the authors add additional fully connected layers as Behavior Module to encode the agent behavior and add the Discriminator to transfer information between behavior modules. The authors experiment on four different games and evaluate based on metrics game scores and behavior distance.

Overall the quality of the paper is low and I recommend to reject it.

[Weakness in Details]
1. I am not convinced that the proposed algorithm actually solves/works as described in the motivation. Moreover, the whole framework just adopts existing algorithms(like DQN and adversarial training) which provides little technical contribution.

2. I am skeptical about the motivation whether mimicking the human brain Pre-Frontal Cortex connectivity can really result in a better neural network model. The poor execution and insufficient evaluation of this work prevent me from getting a clear answer.

3. It is very strange that the authors emphasize that "This property is particularly useful for user modeling (as for dialog agents) and recommendation tasks, as allows learning personalized representations of different user states." while in the end doing experiments with video games playing. There are tons of public recommendation data sets out there, why not experiment on recommendation, which has much clear(well-established) evaluation metrics and public-domain datasets that can make it easier for others to repeat the experiments.

4. The experiments are insufficient and the baselines are weak. Lots of state of artworks are left out.

5. The writing of this paper needs further improvement and parts of this paper is not clearly written which makes it challenging for readers to follow the authors' ideas.

---

### Official Review · AnonReviewer3 · 2018-11-03

**Rating:** 3
**Confidence:** 4

**Review:**

# Summary
This paper proposes to learn behaviors independently from the main task. The main idea is to train a behavior classifier and use domain-adversarial training idea to make the features invariant to sources of behaviors for transfer learning to new behaviors/tasks. The results on Atari games show that the proposed idea learns new behavior more quickly than the baseline approaches.

[Cons]
- Some descriptions are ambiguous, which makes it hard to understand the core idea and goal of this paper.
- The experimental setup is not well-designed to show the benefit of the idea.

# Comments
- This overall idea is a straightforward extension from domain-adversarial learning except that this paper considers transfer learning in RL.
- The goal/motivation of this paper is not very clearly described. It seems like there is a "main task" (e.g., maximizing scores in Atari games) and "behavior modules" (e.g., specific action sequences). It is unclear whether the goal of this paper is to learn 1) the main task, 2) learning new behavior modules quickly, or 3) learning new (main) tasks quickly. In the abstract/introduction, the paper seems to address 3), whereas the actual experimental result aims to solve 2). The term "task" in this paper often refers to "main task" or "behavior" interchangeably, which makes it hard to understand what the paper is trying to do.
- The experiment is not well-designed. If the main focus of the paper is "transfer to new tasks", Atari is a not a good domain because the main task is fixed. Also, behavior modules are just "hand-crafted" sequences of actions. Transfer learning across different behaviors are not interesting unless they are "discovered" in an unsupervised fashion.
- The paper claims that "zero-shot" transfer is one of the main contributions. Zero-shot learning by definition does not require any additional learning. However, they "trained" the network on the new behavior modules (only the main network is fixed), which is no longer "zero-shot" learning.

---

### Meta-Review · Area_Chair1 · 2018-12-17
**Interesting idea that requires more work**

**Confidence:** 5
**Recommendation:** Reject

**Metareview:**

This paper takes inspiration from the brain to add a behavioral module to a deep reinforcement learning architecture. Unfortunately, the paper's structure and execution lacks clarity and requires a lot more work: as noted by reviewers, the link link between motivation and experiments is too fuzzy and their execution is not convincing.